# Clinical Aspects of B Cell Immunodeficiencies: The Past, the Present and the Future

**DOI:** 10.3390/cells11213353

**Published:** 2022-10-24

**Authors:** Aisha Ahmed, Elizabeth Lippner, Aaruni Khanolkar

**Affiliations:** 1Division of Allergy and Immunology, Ann and Robert H. Lurie Children’s Hospital of Chicago, Chicago, IL 60611, USA; 2Department of Pediatrics, Northwestern University, Chicago, IL 60611, USA; 3Department of Pathology, Ann and Robert H. Lurie Children’s Hospital of Chicago, Chicago, IL 60611, USA; 4Department of Pathology, Northwestern University, Chicago, IL 60611, USA

**Keywords:** B cells, immunodeficiency, humoral immunity, inborn errors of immunity

## Abstract

B cells and antibodies are indispensable for host immunity. Our understanding of the mechanistic processes that underpin how B cells operate has left an indelible mark on the field of clinical pathology, and recently has also dramatically reshaped the therapeutic landscape of diseases that were once considered incurable. Evaluating patients with primary immunodeficiency diseases (PID)/inborn errors of immunity (IEI) that primarily affect B cells, offers us an opportunity to further our understanding of how B cells develop, mature, function and, in certain instances, cause further disease. In this review we provide a brief compendium of IEI that principally affect B cells at defined stages of their developmental pathway, and also attempt to offer some educated viewpoints on how the management of these disorders could evolve over the years.

## 1. Introduction

The anamnestic immune response has evolved to provide protective immunity against both extracellular and intracellular pathogens [1]. Two distinct cellular subsets, the B cells (which target the extracellular pathogens) and the T cells (which eliminate intracellular pathogens), serve complementary roles to eradicate the invading pathogens and generate sterilizing immunity [1]. The principal molecule involved in B cell-associated immune responses is the B cell receptor (BCR) or the immunoglobulin (Ig) molecule [1]. As our understanding of how the Ig molecule engages with its cognate antigen and triggers the B cell response has advanced, we have been able to harness this knowledge and revolutionize two major domains of modern healthcare: laboratory diagnostics and immunotherapy. Indeed, it would not be an exaggeration to state that laboratory medicine would still exist in a state of arrested development had Kohler and Milstein not discovered the process to generate monoclonal antibodies (MAb) in the early to mid-1970s [2]. In the ensuing decades, this seminal discovery has also paved the way for the profusion of Ab-based biologic therapies that are now considered the standard of care for several diseases [3]. Similarly, studying patients that suffer from inborn errors of B cell immunity has facilitated the discovery of several key molecular components that regulate B cell development and responses [4]. As a matter of fact, the very first inborn error of immunity (IEI) identified in the human population was a B cell immunodeficiency (X-linked Agammaglobulinemia; XLA) [5]. In this review, we highlight how our understanding of B cell development and humoral immunity has evolved over the past 70 years, based on evaluating the clinical presentations of and laboratory findings in patients with predominantly B cell IEI. We begin this report by providing a brief historical note on XLA, the prototypical B cell IEI, and follow this up with the current state of knowledge relating to a few representative B cell IEI that are routinely considered as part of the differential diagnosis during the clinical workup of patients with B cell and antibody deficiencies. We conclude the report by sharing some thoughts on how diagnosis and the way we treat B cell IEI might be impacted in the future, given that immunoglobulin replacement therapy and prolonged antibiotic treatment may not be a panacea for individuals born with B cell defects.

## 2. The Past

XLA was the first IEI identified in the human population [5]. In 1952, Colonel Ogden Bruton reported a case of an eight-year-old old male child who experienced multiple episodes of bacterial sepsis in early childhood; the laboratory evaluation of his serum sample by protein electrophoresis revealed the absence of the globulin fraction [5]. It took forty years after this seminal observation to pinpoint the exact molecular defect that contributed to this agammaglobulinemia. The affected gene is located on the long arm of the X-chromosome and encodes a cytoplasmic tyrosine kinase that is named in honor of Dr. Bruton, Bruton’s tyrosine kinase (BTK) [6,7]. This disorder affects roughly 1 in 200,000 males and is classically diagnosed when maternal antibodies begin to wane, and the absence of the endogenous humoral immune response renders the child susceptible to childhood infectious diseases, despite receiving childhood vaccines. BTK is present in several cell types that originate from hematopoietic progenitor stem cells, but what differentiates B cells from the other BTK harboring cell lineages is that the signals transduced with the help of this molecule play a critical role in the generation of naive, mature B cells that enter the circulation from the bone marrow. However, it should be pointed out that BTK expression is dispensable in terminally differentiated B cells, as plasma cells lack BTK expression [8]. Consequently, in the absence of BTK expression or if its function is subverted as a result of a crippling mutation, developing B cells in the bone marrow display a maturation arrest and fail to differentiate beyond the pre-B-I cell stage [9]. Therefore, the circulating lymphocyte pool is devoid of B cells, and the patient is rendered agammaglobulinemic. In contrast, the maturation of other cell types of the hematopoietic lineage such as monocytes continues uninterrupted in the absence of BTK expression and/or function, and these cells can successfully egress out of the bone-marrow and seed the circulatory compartment, although some of their effector functions are usually compromised as a result of this defect [10,11]. In the diagnostic laboratory, we exploit this dichotomy in the requirement of BTK relating to the maturation of B cells and monocytes, by assessing the presence or absence of BTK expression in monocytes by flow cytometry, to rapidly rule in or out deficient BTK expression as a potential contributor to the agammaglobulinemic phenotype in a patient (Figure 1). XLA carrier status can also be rapidly ascertained by flow cytometry, as monocytes display bimodal distribution of BTK expression, a consequence of random inactivation (lyonization) of the X chromosome (Figure 1). Although lyonization is also observed in B cells of XLA carriers, the B cells expressing the mutated *BTK* gene are unable to egress out of the bone marrow, hence circulating B cells in XLA carriers always express the wild-type *BTK* gene (Figure 1).

## 3. The Present

As the field of primary B cell IEI rapidly expands, additional patients are being identified who previously had an unclear molecular diagnosis. There are currently nearly 50 distinct clinical entities that fall under the umbrella of predominantly B cell IEI [12,13,14,15] (Figure 2). Discussing each one of these clinical entities is beyond the scope of this report; however, we have highlighted some of the disorders, including a few that have been quite recently identified (PU.1, and disorders of the Ikaros family of transcription factors), which, although uncommon, highlight key steps in the B cell developmental and differentiation pathways (Figure 3).

### 3.1. PU.1

Although XLA is the most common form of agammaglobulinemia, several other congenital agammaglobulinemias have been identified that are inherited in an autosomal dominant or recessive manner (μ chain, Igα, Igβ, λ5, E47, *BLNK*, *PIK3R1*) [4]. One recently discovered cause of congenital agammaglobulinemia is PU.1 haploinsufficiency. Le Coz et al. described six unrelated agammaglobulinemia patients with a heterozygous mutation in *SPI1*, the gene encoding PU.1 [16]. These patients had decreased peripheral B cells and few conventional dendritic cells. PU.1 is a pioneer transcription factor that decompacts stem cell heterochromatin and allows for access to euchromatic regions. Patients with PU.1-related agammaglobulinemia typically present in the first year of life, and have recurrent sinopulmonary, invasive bacterial, and systemic enteroviral infections. Of note, despite the quantitative and qualitative conventional dendritic cell deficiency, these patients do not appear to have issues with mycobacterial infections [16]. PU.1-related B cell immunodeficiency is an excellent example of an early-stage defect, resulting in arrest at the Pro-B cell stage (Figure 3).

### 3.2. Disorders of the Ikaros Family of Transcription Factors

#### 3.2.1. Ikaros Zinc Finger 1 (IKZF1; Ikaros)

The Ikaros family of proteins, including IKZF1 (Ikaros), IKZF2 (Helios), and IKZF3 (Aiolos), are hematopoietic zinc finger DNA-binding transcription factors that regulate lymphocyte development [17]. An IEI that affects B cells in the later stages of development results from mutations in *IKZF1* [18] (Figure 3). Earlier reports have described somatic *IKZF1* pathogenic variants in the context of high-risk B-cell acute lymphoblastic leukemia (B-ALL) development, but more recently our understanding of the *IKZF1*-linked spectrum of diseases has expanded to include several IEI arising from germline *IKZF1* variants [18]. Although these genetic defects are incompletely penetrant, the clinical phenotype in symptomatic patients includes heightened susceptibility to bacterial infections and less commonly, to viral infections, as well as autoimmunity, immune dysregulation and possible increased risk of malignancy (B-ALL) [18]. Dominant-negative variants cause a more severe, combined immunodeficiency associated with pneumocystis pneumonia and T-cell ALL [18]. In contrast, haploinsufficiency-associated variants located at the N-terminus result in a common variable immunodeficiency (CVID)-like phenotype, while variants affecting the C-terminus impair the dimerization of Ikaros and induce hematologic disorders with cytopenias and increased risk of malignancy [18,19]. Recently, a gain-of-function variant in the DNA-binding domain has been reported in eight individuals with inflammatory, allergic and autoimmune manifestations with aberrant T-cell differentiation featuring reduced numbers of regulatory T cells, increased numbers of Th2 cells, peripheral eosinophilia, and abnormal plasma cells [20].

#### 3.2.2. Ikaros Zinc Finger 2 (IKZF2; Helios) and Ikaros Zinc Finger 3 (IKZF3; Aiolos)

IKZF2 is highly expressed in CD8 T cells; consequently, individuals with *IKZF2* pathogenic variants feature impaired T cell homeostasis and can present with an immunedysregulatory/autoimmune phenotype or a combined immunodeficiency phenotype [21]. The clinical presentation can include hypogammaglobulinemia, increased susceptibility to viral and bacterial respiratory infections, thrush, immune thrombocytopenia, systemic lupus erythematosus (SLE), and impaired T-cell activation leading to chronic lymphadenopathy [21,22]. In contrast to IKZF2-related defects, impairments in B cell maturation have been prominently noted in two families with mutations in *IKZF3*. Increased susceptibility to the Epstein–Barr virus (EBV), sinopulmonary infections, marked reduction in B cell numbers, and aberrant T cell development with increased memory and activated T cell subsets, were observed in one family with a heterozygous missense variant in *IKZF3* [23]. A different heterozygous variant was identified in another family with a combined B and T cell deficiency phenotype, *Pneumocystis jirovecii* pneumonia, and chronic lymphocytic leukemia [24].

### 3.3. Hyper IgM Syndromes

One of the more well-known IEI that affects B cell development is a group of disorders classified under the umbrella term of hyper IgM syndromes (HIGM). X-linked HIGM syndrome due to defects in the CD40 ligand (CD40L) account for 65–70% of all hyper IgM cases [25]. Since the identification of CD40 ligand gene (*CD40LG*) mutations resulting in HIGM in 1992, additional genes have been identified that result in a hyper IgM phenotype, where the IgM concentration is normal or increased, and the serum IgG, IgA and IgE levels are low or absent [25]. This group of disorders is also described as immunoglobulin class-switch recombination (Ig-CSR) deficiencies [25]. This nomenclature indicates that the principal underlying defect in these disease processes occurs due to the naïve mature B cell’s inability to isotype switch. Although additional genes have been identified in non-X-linked HIGM syndromes, in this report we will briefly touch upon CD40, activation-induced cytidine deaminase (AID) and uracil-DNA glycosylase (UNG) deficiency disorders.

#### 3.3.1. CD40L (and CD40) Deficiency

CD40L and CD40 defects result in both humoral and cellular immunodeficiency, due to the important role of CD40L/CD40 interaction in T cell co-stimulation. Clinically, CD40 deficiency is similar to CD40L deficiency, especially with regards to pathogen susceptibility [25]. The main clinical features of CD40L defect are early onset of infectious complications, recurrent sinopulmonary infections, mainly due to encapsulated bacteria, and susceptibility to opportunistic infections caused by Pneumocystis, Cryptosporidium, Mycobacteria, Cytomegalovirus (CMV), Toxoplasma and Cryptococcus [25,26,27,28,29,30,31,32,33,34]. Cryptosporidium parvum-related biliary disease can be fatal, resulting in sclerosing cholangitis or cholangiocarcinoma [25]. Additional features include oral ulcers, immune-mediated cytopenias, osteopenia, and increased risk of malignancy [35,36].

#### 3.3.2. AID

AID (encoded by *AICDA*) plays an important role in class-switch recombination and somatic hypermutation [25]. Patients with AID deficiency typically become symptomatic in the first decade of life but can also present after the second or third decade of life [25]. These patients are typically susceptible to infections with encapsulated bacteria, but other pathogens have also been identified [25]. Autoimmunity and lymphoid hyperplasia are non-infectious complications noted in patients with AID deficiency [25].

#### 3.3.3. UNG

UNG is an enzyme that is also important for class-switch recombination and somatic hypermutation [25]. Its enzymatic activity is an important step, which follows AID-induced deamination [25]. UNG deficiency is rarer than AID but has an identical clinical phenotype [25].

### 3.4. Common Variable Immunodeficiency (CVID) and CVID-like Disorders

#### 3.4.1. CVID

Among the defects beyond the class-switching phase of B cell development and differentiation, CVID is the most common disorder. The first clinical description of CVID was published in 1954, and it described an adult female with agammaglobulinemia, recurrent infections including bacterial pneumonias, and additional chronic lung disease and diarrhea [37]. Since that time, the general perception of CVID has been that it is a disorder of adults with recurrent and prolonged infections and hypogammaglobulinemia. However, data from U.S. and European registries indicate that pediatric cases account for 18–42% of CVID cases [38,39]. CVID is purported to affect between two and four individuals per 100,000 people, and there is no significant predilection for any single ethnicity or gender. Following diagnosis, initiation of immunoglobulin replacement therapy, via intravenous or subcutaneous administration, along with infection prevention strategies and early intervention for bacterial infections, are the mainstays of treatment.

As there is no single clinical or laboratory finding that is pathognomonic for CVID, several diagnostic criteria have been proposed [40,41,42]. While these criteria vary slightly, the accepted criteria according to the International Consensus Document (ICON) include immunodeficiency symptom onset at two years of age or older, an immunoglobulin G (IgG) level more than two standard deviations below the mean based on age, plus low immunoglobulin A and/or low immunoglobulin M level(s), and poor Ab responses to vaccines [42]. It is also contingent upon exclusion of other potential causes/disorders such as immunosuppression, malignancy, or another genetic condition [42]. Definitive evidence of hypogammaglobulinemia consists of IgG levels measured twice, at least three weeks apart. Patients with profound laboratory abnormalities, such as an IgG level less than 100 mg/dL, should be considered for CVID and other B-cell IEI, even if the patient lacks a significant infectious or autoimmune history. These individuals are at risk of developing infectious or autoimmune complications over time, as the laboratory criteria may precede development of clinical symptoms. Similarly, while infections may begin in childhood, the patient may not meet full diagnostic criteria for several years. The time frame from onset of symptoms to diagnosis has been reported to be 5–6 years in US cohorts and 7.5 years in European cohorts [42]. Although not an official criterion, low numbers of isotype-switched memory B cells (<2%) are also commonly observed and are supportive of the diagnosis.

CVID is inherently a diagnosis of exclusion. Ruling out other etiologies can be particularly challenging in pediatric patients, as many IEI present in childhood, children often contract viral illnesses that are treated as bacterial infections, and young children may not have received all childhood vaccines at the time of initial evaluation. As such, children initially diagnosed with CVID under four years of age often require periodic re-evaluation of this diagnosis.

As part of the evaluation for CVID, it is necessary to define what constitutes “poor vaccine responses”. The consensus diagnostic criteria include assessment of both T-dependent and T-independent vaccine responses. Several T-dependent vaccines have established values at which one is considered protected or responsive, such as tetanus (0.15 IU/mL) and *Haemophilus Influenzae* type B (1 µg/mL) [43]. The most common T-independent vaccine is the 23-valent pneumococcal vaccine, Pneumovax [44]. Accurate assessment of one’s response to Pneumovax requires measurement of titers at baseline and then again at four to eight weeks post-vaccination [45]. While a level of 1.3 μg/mL or a four-fold increase in the titer are generally accepted as criteria for an appropriate response, there is variability across populations regarding the response to specific *Streptococcus pneumoniae* (*S. pneumoniae.*) strains included in the vaccine [46]. Protective responses to 70% of the *S. pneumoniae.* strains in those 6 years of age and older and 50% of the *S. pneumoniae*. strains in children 2–5 years of age, are used to define normal polysaccharide vaccine responses; however, clinicians should also consider the individual’s clinical history [45].

While the most recognized clinical feature of CVID and many IEI is recurrent or severe infection, there is a growing body of literature elucidating the non-infectious symptoms of many IEI, CVID included. Non-infectious complications are of significant consequence, as they can affect both quality of life and long-term prognosis/outcomes. Ho et al. found that among 623 patients with CVID followed in New York state from 1974 onwards, 68% had non-infectious manifestations, including 33.25% with an autoimmune complication [47]. Similarly, Odnoletkova et al. found that CVID patients on the European Society for Immunodeficiencies registry had a similar proportion with autoimmune manifestations [48]. Immune-mediated cytopenias, more specifically Evan’s syndrome and Immune thrombocytopenic purpura (ITP), are the most reported autoimmune manifestations [48]. Other autoimmune complications in CVID include arthritis, autoimmune thyroid disease, pernicious anemia, pancreatitis, myasthenia gravis, SLE, antiphospholipid antibody syndrome, vasculitis, type 1 diabetes, lichen planus and vitiligo [47]. Autoimmune manifestations may pre-date infections and be the presenting symptom of CVID, a fact that may delay the diagnosis of CVID if a thorough clinical history is not obtained and relevant laboratory testing is not performed. Ho et al. additionally reported that a substantial percentage of patients had other non-infectious complications, including: (1) chronic lung disease, with bronchiectasis and interstitial lung disease being the most prominent, and affecting 30.3% of patients; (2) gastrointestinal and bowel disease, characterized by villous atrophy, intraepithelial lymphocytosis, and absence of plasma cells, among other histopathologic findings, and affecting roughly 17% of patients; (3) liver disease, characterized by granulomas and/or nodular regenerative hyperplasia, affecting 12.7% of patients; (4) lymphoproliferative disease such as lymphoid hyperplasia and/or splenomegaly, affecting 20.9%; and (5) lymphoma, affecting 6.7% [47]. Immunoglobulin replacement therapy, while providing protection from infection, is not therapeutic for these non-infectious complications, nor does it prevent development of these non-infectious complications. Moreover, Resnick et al. found that CVID patients with any non-infectious complication have an 11-fold higher risk of death compared with those with CVID and only infectious complications [49]. Mortality was strongly associated with the presence of chronic lung disease, lymphoma, hepatitis, and gastrointestinal disease, but not with autoimmunity or bronchiectasis [49].

CVID was historically understood as a single disorder; however, the significance of these non-infectious complications serves to amplify the fact that CVID features a wide constellation of clinical symptoms which varies among individuals. Additionally, the application of genetic sequencing in recent decades has allowed for reclassification of a subset of patients previously diagnosed with CVID based on consensus criteria, but who were later found to have an identifiable monogenic cause of disease or variants in genes associated with poor B cell maturation and survival (e.g., Transmembrane Activator and CAML Interactor [TACI]. These findings have prompted a redefining of CVID, not as a single diagnosis but rather as a phenotype that encompasses several hypogammaglobulinemia syndromes. In 2009, the International Union of Immunologic Societies (IUIS) Expert Primary Immunodeficiency Committee renamed the condition “common variable immunodeficiency disorders”, to better reflect these heterogenous syndromes [50]. This is also reflected in the categorization/tables of the IUIS inborn errors of immunity/primary immunodeficiencies [12,13].

#### 3.4.2. Variants in the TNF Receptor Superfamily Member 13B (*TNFRSF13B*)

The *TNFRSF13B* gene codes for the protein TACI. TACI is expressed on both B cells and activated CD4 T cells and participates in Ab isotype switching by interacting with signaling ligands BAFF and APRIL, to affect B cell maturation [51]. TACI is also believed to play a role in B cell activation and removal of auto-reactive B cells [51]. Variants in the *TNFRSF13B* gene have been identified in 8–10% of individuals with CVID, although *TNFRSF13B* variants are also commonly found in the unaffected population [51]. Therefore, *TNFRSF13B* variants may confer increased susceptibility to CVID and IgA deficiency, but they are not a putative cause of CVID.

#### 3.4.3. Mutations in *NFKB1*

One of the most common causes of CVID is heterozygous mutations in *NFKB1* which leads to haploinsufficiency of the NFκB p50 subunit. First described in 2015, we now have an expanded understanding of the diverse clinical phenotype that arises due to dysregulated NFκB signaling. Interestingly, there is an age-dependent penetrance which may explain the disease evolution that has been noted in many patients [52,53]. Complications can include hypogammaglobulinemia and infections affecting the respiratory and GI tract, as well as opportunistic infections. Non-infectious complications can include autoimmunity, autoinflammation, lymphoproliferation, malignancy and enteropathy [53,54].

#### 3.4.4. Cytotoxic T-Lymphocyte-Associated Protein-4 (CTLA-4) Insufficiency

CTLA-4 functions as an immune checkpoint/negative regulator of T cell effector function. It is constitutively expressed on T regulatory cells and also expressed on activated effector T cells. The inhibitory function of CTLA-4 is mediated through its interaction with CD80 and CD86 (B7-1 and B7-2) expressed on antigen-presenting cells (APC). This binding leads to the removal of CD80/CD86 from the APC surface (trans-endocytosis), thereby preventing the interaction between CD80, CD86 on the APC with CD28 on the T cells, resulting in the inhibition of effector T cell function [55]. In CTLA-4 insufficiency, there is unchecked T cell activation with subsequent development of lymphoproliferation and escape of autoreactive T cells [56,57]. The plethora of clinical manifestations include recurrent infections, hypogammaglobulinemia, lymphoproliferation (diffuse lymphadenopathy, hepatosplenomegaly, and lymphocytic infiltration of nonlymphoid organs), cytopenia, and lymphopenia with decreased but variable levels of T, B, and NK cell counts. There is also increased risk of non-Hodgkin’s lymphoma and EBV-associated malignancies, demonstrated in a cohort study where six of nine patients with either lymphoma or gastric cancer had an EBV-associated malignancy [56,57,58]. Lymphocytic infiltration involving the lungs, liver, intestines, and CNS have been reported with significant end-organ damage [58,59,60]. *CTLA4* mutations are autosomal dominant but there is incomplete penetrance with a variable clinical phenotype. Moreover, family members with identical pathogenic mutations may not share the same clinical symptoms/phenotype. Given the varied clinical symptoms, multiple treatment modalities may be employed, including immunoglobulin replacement therapy, aggressive infectious management, the CTLA-4-Fc fusion protein (abatacept) has been used as therapy for both adult and pediatric patients, rituximab, Sirolimus, systemic corticosteroids, and hematopoietic stem cell transplantation (HSCT) [61]. In a cohort review of 173 individuals with *CTLA4* mutations, abatacept, rituximab, sirolimus, and steroids were reported as modifying disease severity, and 13 of 18 patients that received an HSCT were alive and well without immunomodulatory medications with a median follow up time of 20.8 months post-HSCT [60]. 

#### 3.4.5. LPS Responsive Beige-like Anchor Protein (LRBA) Deficiency

LRBA is a cytosolic protein expressed in all cells that functions in vesicle trafficking and regulates endosomal trafficking of Cytotoxic T Lymphocyte-Associated Protein-4 (CTLA-4) [62]. LRBA deficiency impairs endosomal recycling and surface expression of CTLA-4. Patients with autosomal recessive mutations in *LRBA* suffer from recurrent infections with a CVID-like phenotype featuring prominent autoimmune complications such as inflammatory bowel disease, psoriasis, autoimmune cytopenias, thyroid disease, and diabetes [62]. Immunophenotyping of patients with LRBA deficiency reveals decreased numbers of regulatory T cells (Tregs), isotype-switched memory B cells, and plasmablasts, in addition to decreased immunoglobulins and decreased CTLA-4 expression [62]. While patients with LRBA deficiency can have a similar phenotype to CTLA-4 insufficiency, LRBA deficient patients tend to have high rates of splenomegaly, double-negative T cells and low numbers of Tregs, and decreased plasmablasts [63]. Additionally, there are reported cases of LRBA deficiency and early onset of type I diabetes [63].

#### 3.4.6. Activated Phosphoinositide 3-Kinase Delta Syndrome (APDS)-1/2 (Defect in PIK3CD/PIK3R1)

APDS-1 arises from autosomal dominant gain-of-function mutations in the *PIK3CD* gene that encodes for the catalytic (p110δ) subunit of phophotidylinositol 3-kinase (PI3K), while monoallelic loss-of-function mutations in *PIK3R1*, which encodes for the regulatory (p85α) subunit of PI3K, causes APDS-2 [64]. Both conditions result in a similar clinical phenotype, characterized by recurrent infections with a CVID-like or hyper-IgM-like phenotype, and increased susceptibility to CMV and EBV infections, as well as an increased risk for B cell lymphomas [64]. Symptom onset is typically in childhood, and autoimmune cytopenias are commonly observed. The characteristic immunophenotype includes variable serum Ig levels (ranging from hypogammaglobulinemia to hypergammaglobulimemia to those with a hyper IgM phenotype), reduced Ag-specific antibody titers, peripheral T and B cell subset lymphopenia with inverted CD4/CD8 ratios, low numbers of isotype-switched memory B cells, and evidence of impaired cytotoxic potential of CD8+ T cells and NK cells [64]. Downstream effects of APDS-1 cause overactivation of the mammalian target of rapamycin (mTOR) pathway [64]. The potent mTOR inhibitor, sirolimus (rapamycin), is a directed therapeutic option in activated PIK3CD/APDS [64]. Clinical trials are also underway, to examine the therapeutic efficacy of leniolisib, a selective PI3K δ inhibitor [65].

## 4. The Future

### 4.1. The Role of Kappa (Light Chain Gene) Rearrangement Excision Circle (KREC) Quantification, Serum Biomarkers and Gene Sequencing in the Diagnosis of B Cell IEI

The future of B cell deficiencies includes a focus on early detection and a trend towards molecular precision treatment, away from a standard approach to all patients. As discussed in the previous sections, the diagnosis of primary B cell deficiencies can be challenging and cumbersome. Currently, children with predominantly B cell disorders are usually diagnosed after maternal Abs wane. Maternal Abs (transplacentally-acquired IgG and breast milk-derived secretory IgA) provide critical but temporary immune support during the neonatal period and early infancy, while the child’s own immune response is getting primed and maturing, following exposure to environmental and vaccine-derived antigens. However, in children born with numerical and/or functional insufficiency of the B cell compartment, the lack of an endogenous humoral immune response renders them highly susceptible to a host of pathogens, and these individuals can eventually experience debilitating conditions such as chronic lung disease and bronchiectasis, despite receiving life-long IgG replacement therapy [66,67]. In addition, current diagnostic modalities require temporary cessation of immunoglobulin replacement therapy which may take several months to complete, which puts patients at risk of severe infection. Newer laboratory methods focused on the timely identification of B cell deficiencies have been developed, and novel serum biomarkers have been proposed. One such promising tool is the KREC assay [68]. One of the biggest success stories in the world of clinical immunology over the past two decades has been the implementation of the severe combined immunodeficiency (SCID) newborn screen, utilizing a T cell receptor rearrangement excision circle (TREC) assay [68]. Early detection of SCID has facilitated early treatment, and directly led to decreased mortality in this cohort of patients [69]. The success of the TREC assay has encouraged the consideration of utilizing similar techniques to identify B cell disorders through newborn screening. Just as T cells undergo the process of V(D)J recombination to produce unique T cell receptors (a process that yields TRECs), B cells also produce unique B cell antigen receptors, and generate KRECs [68]. Kappa (κ) light chain gene rearrangement occurs prior to the lambda (λ) light chain gene rearrangement in pre-B cells, thus every mature B cell, irrespective of whether it ultimately expresses the κ light chain or the λ light chain, harbors KRECs [68]. Implementation of the KREC assay has been proposed in various clinical scenarios, and in 2011 Nakagawa et al. first demonstrated that B cell deficiencies could be identified based on decreased KREC levels [70]. Over the last ten years, additional studies have evaluated the role of the KREC assay alone or, more commonly, multiplexed with the TREC assay, to identify primary immunodeficiencies that involve early B cell defects [71,72,73]. Although KREC quantification appears to be successful based on pilot studies from various countries such as Switzerland and Iran, full public health benefits remain to be established [71,72,73,74]. Another approach towards minimizing the delay in identification of primary B cell deficiencies involves the use of serum biomarkers that can play both a diagnostic and a prognostic role. Several potential candidates have been proposed, but perhaps the most promising is the B cell maturation antigen (BCMA) [75,76,77,78]. BCMA is a member of the tumor necrosis factor receptor superfamily, and it promotes the survival of plasma cells and plasmablasts [78]. It is proving to be an exciting therapeutic target for multiple myeloma, and recently attention has turned to its utility in diseases at the other end of the B cell proliferation spectrum [78]. Maglione et al. measured serum BCMA levels in 165 patients and discovered that serum BCMA was significantly reduced in severe primary antibody deficiencies such as XLA and CVID, compared with less profound antibody deficiencies such as selective IgA or mild IgG deficiency [78]. Although larger scale prospective studies are required to validate the utility of serum BCMA levels, the concept of utilizing serum biomarkers in identifying primary B cell deficiencies remains enticing. The goal of both newborn screening and identification of serum biomarkers is to reduce the delay in diagnosis of primary antibody deficiencies which can lead to irreversible end-organ damage [79].

Along with identification of laboratory markers, the future of primary B cell disorders relies heavily on the utilization of genetic testing modalities. In 2016, Maffucci et al. published a study in which whole exome sequencing (WES) combined with analysis of PID-associated genes was performed on 50 patients diagnosed with CVID [80]. At that time, 269 primary immunodeficiency-causing genes had been identified, and this number has nearly doubled in the last five years [12,13]. Maffucci et al. reported that their approach including WES combined with analysis of PID-associated genes was cost-effective and successful in identifying disease-causing mutations in 30% of CVID patients with severe phenotypes [80]. However, an intriguing, yet unsurprising result from this study was the discovery of monoallelic and biallelic mutations that cause more severe primary B cell deficiency, such as *NFKB1*, *STAT3*, *CTLA4*, *PIK3CD*, *IKZF1*, *LRBA* and *STXBP2* [80]. Since then, additional literature has emerged regarding the benefits of genetic testing of patients with a severe CVID phenotype, as it can uncover a more precise diagnosis in which the hypogammaglobulinemia may just be one piece of the clinical picture [81]. The push towards a more specific molecular diagnosis is due to the advent of targeted therapy that takes a precision medicine approach to the treatment of primary B cell deficiencies [82,83]. For example, instead of using broad immunosuppression such as systemic corticosteroids to manage cytopenias and lymphoproliferation in patients with activated PIK3CD (APDS-1), a targeted approach such as leniolisib (a selective PI3K δ inhibitor) may be beneficial and have decreased toxicity [65]. Harnessing the power of genetic testing and utilizing those results to guide targeted therapies is a central component of the future management of primary B cell IEI.

### 4.2. Allogeneic Hematopoietic Stem Cell Transplantation (HSCT)

Another approach to the treatment of primary B cell deficiencies includes allogenic HSCT as a curative therapy [84]. The overarching paradigm thus far has been that the defect in the humoral immune compartment can be addressed with immunoglobulin replacement therapy +/− antimicrobial prophylaxis [85,86]. However, as recent case reports have demonstrated, the difficulty in clearing specific viral infections in XLA, such as chronic norovirus and Aichi virus, may require more extensive treatment, such as HSCT [87,88]. But HSCT-associated complications, which can include a high rate of mortality, have tempered the enthusiasm for this approach in patients with humoral insufficiency, although currently it may be the only useful option in a subset of those patients that develop intractable complications such as refractory cytopenias, in spite of long-term immunoglobulin replacement [89].

### 4.3. Therapeutic IgA

Recent data from a long-term follow-up study of 168 XLA patients demonstrated that despite regular Ig replacement therapy, the cumulative risk for developing chronic lung disease (CLD) and bronchiectasis in the study subjects was 47% at 50 years of age [66]. Furthermore, in this study, XLA patients with and without CLD displayed significantly reduced overall survival levels compared with the general population, and this reduction in overall survival was more pronounced if CLD was part of the clinical picture [66]. Gastrointestinal pathology is another prominent clinical manifestation of primary B cell IEI [67]. This is not altogether surprising, given that the gastrointestinal (GI) tract is exposed to a myriad of foreign antigens on a daily basis, which has necessitated the evolution of the intestine as the largest lymphoid organ in the body [67]. Both the respiratory and GI tracts are mucosal tissues where the predominant Ig that mediates robust protection is secretory IgA (sIgA), a heavily glycosylated tetrameric complex comprised of an IgA dimer, the J chain and the secretory component (SC) [90,91]. It is this unique structural assembly that allows sIgA to survive that harsh mucosal environment and discharge its immune effector functions optimally [90,91]. In recent years, it has also become evident that sIgA plays a critical role in regulating the composition of the gut microbial flora, preventing dysbiosis and thereby maintaining gut health [90,91]. Hence, given that IgA constitutes only a minor component of the FDA-approved Ig replacement products, and the fact that Ig replacement products are derived from human plasma/serum, a site where IgA exists as a monomer rather than as sIgA, the resultant GI pathology and the progression to CLD in many patients with B cell IEI, despite lifelong Ig replacement therapy, is not surprising [85,90,91]. Although IgA replacement therapy has been attempted in humans, this approach has been variably effective because not all of these replacement products contained sIgA, and IgA is normally transported from sites of local production (and not the blood stream) to the mucosa [90,92,93,94,95,96,97]. However, there continue to be incremental, albeit critical, advances in IgA biology relating to glyco-engineering, as well as recombinant DNA expression methodologies that raise the possibility of sIgA replacement therapy becoming part of the standard of care for B cell IEI patients in the future [91,98,99,100]. Some of the scientific bottlenecks in this regard, for which solutions are being actively pursued, include extending the half-life of IgA by engineering its association with the neonatal Fc receptor and addressing the feasibility of commercial scale production by generating sIgA in various eukaryotic expression systems, including plants, for oral consumption by humans [91,98,100]. Feasibility studies are also being conducted, using polymerized human plasma-derived IgA, and coupling them to recombinant SC to generate sIgA molecules, which are both protease-resistant and fully functional in the harsh mucosal environment in vivo [98]. Additional research will also be required, to determine the need to supplement novel Ig replacement products with molecules such as recombinant hyaluronidase, which might facilitate tissue uptake, similar to what is currently available with supplemental IgG formulations [85,101]. If and when therapeutic IgA becomes part of the standard of care, in addition to using it to treat patients with overt and frank humoral insufficiency, this approach could also potentially benefit the subset of individuals that do display clinical evidence of disease in the setting of selective IgA deficiency.

### 4.4. Gene Therapy

As mentioned previously, current treatment strategies of immunoglobulin replacement +/− antimicrobial therapy may not be enough, and although allogeneic HSCT offers a definitive cure for disorders affecting cells of the hematopoietic compartment, allogeneic HLA-matched donors are not available for almost two-thirds of transplant patients [102]. Furthermore, while the success rate for allogeneic HSCT is almost 90% for patients where the donor is an HLA-matched sibling or family member, this statistic falls to less than 70% if the HLA-matched donor is unrelated, and plummets to less than 50% where the donor is a haploidentical sibling, due to complications such as graft-versus-host disease (GvHD) and immunosuppression-mediated life-threatening infections [102]. In these instances gene therapy can be a potentially reasonable alternative and may be the just right approach [103]. Over the past two decades gene therapy has been successfully implemented for some monogenic IEI such as adenosine deaminase deficiency and X-linked severe combined immunodeficiency [102,103,104]. This therapeutic approach entails the ex vivo transduction of autologous HSC by viral vectors carrying the correct transgene of interest, followed by the adoptive transfer of these transduced HSC with or without pre-conditioning [102,103,104]. Clinical trials over the years have led to marked improvements in the viral vector design, which have enhanced the safety profile of this approach by reducing the risk of leukemia from insertional mutagenesis. The ability to successfully cryopreserve and transport the transduced cells has also advanced gene therapy closer to greater clinical application, and phase I/II clinical trials have also been initiated for other IEI such as chronic granulomatous disease (CGD) and Wiskott–Aldrich Syndrome (WAS) [102,103,104]. However, safety and efficacy concerns persist and need to be systematically addressed for this therapeutic option to become broadly applicable. Furthermore, gene therapy is currently quite cost prohibitive. Hence, viable business models will need to be developed that address cost-effectiveness from the production end (without compromising quality), and affordability, from both the physician and patient standpoint.

### 4.5. Gene Editing

One of the most exciting new developments in translational genomics is the discovery and identification of the process of homology-directed repair of gene defects (gene editing), which offers the tantalizing prospect of rectifying genetic defects in situ [104,105,106]. The potential advantage of this approach is a reduced risk of insertional mutagenesis, with the added benefit of more physiologically appropriate regulation of gene expression [104,105,106]. This field continues to make rapid strides, and within the scientific paradigm of discovery → optimization → deployment (into clinical practice), gene editing currently resides within the optimization realm. There are several pre-clinical studies underway that are addressing critical parameters such as safety, efficacy, timing, and dosage, as well as fine-tuning the intracellular delivery of editing reagents [104,105,106]. Recently, Gray et al. described their experience with editing the *BTK* locus, where they determined that the addition of the *BTK* terminal intron to the donor template led to a significant increase in BTK expression [107]. Based on the results of this study, the authors also surmised that the survival advantage of BTK+ B cells should theoretically allow for even a fraction of edited hematopoietic stem cells to confer a cure of the underlying immunodeficiency [107]. Another crucial aspect of gene editing that will demand attention as it approaches the juncture of clinical testing is the type of cell (hematopoietic stem cells or induced pluripotent stem cells) to utilize to ensure long-term self-renewal capacity of the in vitro manipulated cell population following transfer into the host [104,105,106].

## 5. Conclusions

Evaluating patients with primary B cell IEI has unearthed a treasure trove of fundamental knowledge that undergirds our understanding of B cell biology. Going forward, as we identify more patients with novel primary B cell IEI and gain a better appreciation of the spectrum of clinical phenotypes that can be associated with these disorders, it will be most informative to couple this information with advances in genomics, systems immunology and computational biology. This in turn might potentially enable a re-purposing of our understanding of how B cells operate, in order to design new therapies and cures for diseases, similar to the way in which CAR-T cell therapy (which utilizes ectopic Ab expression in T cells to target tumor antigens) has shifted the paradigm for managing certain malignancies that were once deemed untreatable [108].

## Figures and Tables

**Figure 1 cells-11-03353-f001:**
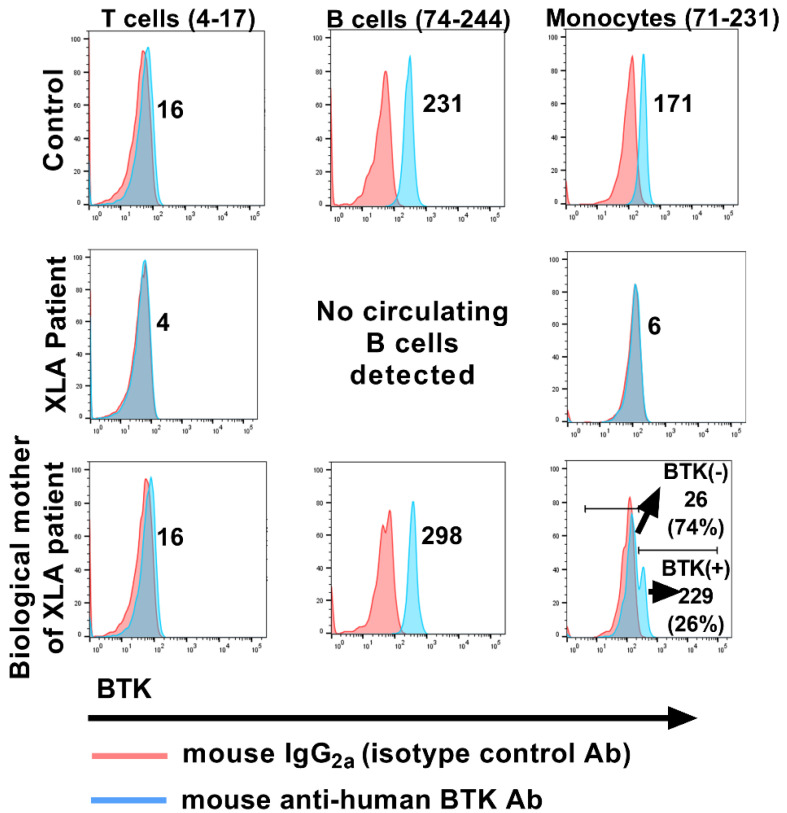
Flow cytometry−based assessment of BTK expression to aid in the clinical diagnosis of XLA. The histogram overlay plots reveal the presence and/or absence of BTK protein expression in the cytosol of the indicated cellular subsets. Circulating T cells, B cells and monocytes were identified by utilizing lineage−directed monoclonal antibodies targeting the surface expression of CD3, CD19 and CD64. Subsequently, cells were fixed, permeabilized and stained with a BTK−specific monoclonal Ab (clone 53/BTK; BD Biosciences, San Jose, CA, USA) to facilitate the detection of cytosolic BTK (blue histograms). The numbers listed in parentheses adjacent to the cell subset label reflect the clinically−validated reference ranges (5th−95th percentiles) for that particular cell subset, derived from the background−subtracted median fluorescence intensity (MFI) of BTK expression. The numbers listed within each histogram overlay plot reflect the background−subtracted MFI of BTK expression for that specific cell subset for each donor. The histogram overlay plots for the carrier monocytes list the background−subtracted BTK MFI values, as well as the frequency (%) of the BTK(+) and BTK(−) monocytic populations. The specificity of the BTK signal was determined by also separately staining the cells with a dose−matched, isotype control Ab (pink histograms). T cells serve as an additional internal specificity control in the assay, as they lack BTK expression.

**Figure 2 cells-11-03353-f002:**
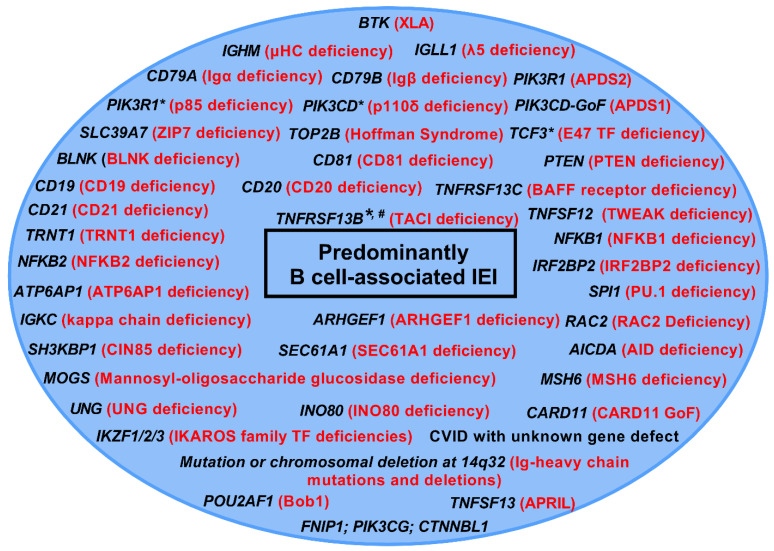
Listing of predominantly B cell associated IEI. This figure lists both the underlying genetic defects (in black) and name of the disease (in red), based on the 2019 International Union of Immunological Societies (IUIS) update, as well as the 2022 update on the classification of human inborn errors of immunity. * These defects can display both autosomal recessive (AR) as well as autosomal dominant (AD) patterns, and these patterns can affect the disease phenotype as well as the degree to which the different Ig isotypes and B cell numbers are decreased. ^#^ These variants can be considered disease−modifying rather than disease−causing, since healthy individuals can also harbor heterozygous variants of this gene. TF: transcription factor.

**Figure 3 cells-11-03353-f003:**
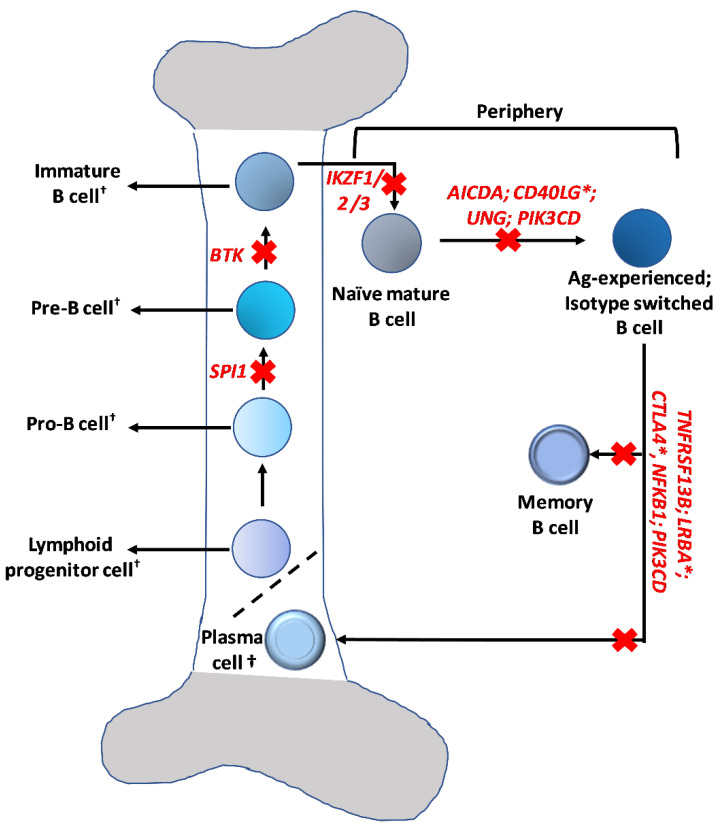
Representative genes targeting defined steps in the B cell maturation and differentiation pathways. Immune defects caused by the genes (labeled in red) are discussed in this report. * Although these genes are not included in the latest IUIS classification of predominantly Ab deficiencies, they are associated with defects in Ab isotype switching (*CD40LG*), and immune dysregulation, manifesting as early−onset hypogammaglobulinemia (CTLA4 and *LRBA*), which is part of a spectrum of widespread immune aberrations that also includes autoimmunity. † These cell populations reside in the bone−marrow.

## Data Availability

Not applicable.

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
