# Peer review of "Clinical Aspects of B Cell Immunodeficiencies: The Past, the Present and the Future"

_cells, 2022, doi:10.3390/cells11213353_

Round 1

Reviewer 1 Report

Overall a good review about an interesting topic,  B cell defects.    The reader does wonder why some rare defects (PU1.1 and Ikaros) are chosen and others are not. What was the rationale?  --   Examples the second most common gene defect in CVID ( NFKB1) is not included – LRBA is included but CLTLA4,  which is more common, is not.

Some alterations and corrections are suggested

Page 75:   Please modify:    "although ( some of ) their effector functions are usually compromised as a result of this defect" [10,11].      More global defects have not been identified.

Line 105  Ref 13 needs an update: 

Tangye SG, Al-Herz W, Bousfiha A, Cunningham-Rundles C, Franco JL, Holland SM, Klein C, Morio T, Oksenhendler E, Picard C, Puel A, Puck J, Seppänen MRJ, Somech R, Su HC, Sullivan KE, Torgerson TR, Meyts I. Human Inborn Errors of Immunity: 2022 Update on the Classification from the International Union of Immunological Societies Expert Committee. J Clin Immunol. 2022 Jun 24:1-35.   

Line 202 “ However, CD40 deficiency is less well understood compared to CD40L  deficiency and is likely under reported.”  Where is the data on that statement?     

Refs 24-28:  adding a collection  of  many cases rather than single case reports might provide a better reference for the reader. 

Line 244 “ Asymptomatic patients without a significant infectious history may meet criteria if they have profound laboratory abnormalities such as an IgG level less than 100 mg/dL.”  No the definition does not include any infectious history at all.    If the patient has low IgG, IgA and or IgM and lacks antibody, and other causes are excluded, it is called “CVID”.   

Line 256 “As such, applying  the diagnosis ( CVID)  to children less than 4 years of age is discouraged.” True, but also it allows the investigator to see if there is just physiologic immaturity or another gene cause.

Line 286  “A response to 70% of the S.p. strains in children and 268 50% of the S.p. strains in adults is generally considered acceptable [36].”     This reference cannot be retrieved; please check it.  And wasn’t there an age consideration attached to this statement?  

Line 317   This paragraph is  headed “ TACI Deficiency”  and is about the TNFRSF13B (TACI)    gene not the TNFSF13B gene (which encodes  BAFF).  It is confusing to have BAFF here as this gene is not something noted as defective in CVID.  

Line 325: BAFF Deficiency ?  this paragraph is not about a lack of BAFF, it only discusses the excess of BAFF.   Please remove as a genetic cause.

Line 474  “Although IgA replacement therapy has been attempted in humans, this approach has been variably effective because not all of these replacement products contained  sIgA”   [69,71-76].  But would IgA or sIgA given by vein, ever be secreted  into the mucosa?  This was a main issue in these articles.  IgA seems to be transported from local production into mucosa, not from the blood stream.

Author Response

We have uploaded a pdf document that includes a point-by-point response to the comments from Reviewer 1

Reviewer 2 Report

Please consider:

- Add CLTA4 deficiency together with LRBA deficiency

- Decrease the enthusiasm for HSCT. Mortality might reach up to 50%, and justified only for those who develop life-thereatening complications.

- Comment on selective IgA deficiency, which is considered common in healthy individuals. TProponents of therapeutic IgA approaches might first identify which subset of patients with selective IgA deficiency do have an increased risk of complications. 

Author Response

We have uploaded a pdf document that includes a point-by-point response to the comments from Reviewer 2.
